# Executive Function among Chilean Shellfish Divers: A Cross-Sectional Study Considering Working and Health Conditions in Artisanal Fishing

**DOI:** 10.3390/ijerph18115923

**Published:** 2021-05-31

**Authors:** Marie Astrid Garrido, Lorenz Mark, Manuel Parra, Dennis Nowak, Katja Radon

**Affiliations:** 1Center for International Health@Occupational, Social and Environmental Medicine, University Hospital Munich, Ludwig-Maximilians-University, 80539 Munich, Germany; lorenzmark99@gmail.com (L.M.); manuel.parra@uda.cl (M.P.); Katja.Radon@med.uni-muenchen.de (K.R.); 2Faculty of Medicine, University of Atacama, Atacama 1532297, Chile; 3Institute for Occupational, Social and Environmental Medicine, University Hospital Munich, Ludwig-Maximilians-University, 80539 Munich, Germany; Dennis.Nowak@med.uni-muenchen.de

**Keywords:** decompression sickness, embolism, air, informal sector, Wisconsin Card Sorting Test, diving, occupational health

## Abstract

Knowledge about professional diving-related risk factors for reduced executive function is limited. We therefore evaluated the association between decompression illness and executive functioning among artisanal divers in southern Chile. The cross-sectional study included 104 male divers and 58 male non-diving fishermen from two fishing communities. Divers self-reported frequency and severity of symptoms of decompression illness. Executive function was evaluated by perseverative responses and perseverative errors in the Wisconsin Card Sorting Test. Age, alcohol consumption, and symptoms of depression were a-priori defined as potential confounders and included in linear regression models. Comparing divers and non-divers, no differences in the executive function were found. Among divers, 75% reported a history of at least mild decompression sickness. Higher frequency and severity of symptoms of decompression illness were associated with reduced executive function. Therefore, intervention strategies for artisanal divers should focus on prevention of decompression illness.

## 1. Introduction

Small-scale and artisanal fisheries are key for food security, poverty eradication, and sustainable development worldwide. Over 90% of the world’s fishing-related workers are engaged in small-scale fisheries [1]. However, precarious working and employment conditions, environmental and socioeconomic factors, and lack of access to social welfare services affect the health of the families and welfare of the communities that directly depend on this sector [1].

In Chile, artisanal fishing takes place mainly in rural coastal communities. Shellfish divers participate in artisanal fishing by harvesting and capturing marine species such as shellfish, crustaceans, mollusks, and algae. They supply raw materials to the local market, to intermediaries or to processing companies that supply the international trade [2]. Nationally, 58% of the 38,440 divers registered at the Maritime Authority in 2017 were basic shellfish divers. In addition, there is an unknown number of unregistered divers [3]. 

Under water, most basic shellfish divers in Chile breathe compressed air by the so-called Hookah technique [3,4]. The air is supplied from an air compressor located on the surface of the boat and passes into the diver’s mouth through a hose and a regulator. While the diver is harvesting, one or two dive assistants maintain the Hookah breathing system and the boat in proper functioning [3]. The assistants also load the products harvested by the diver into the boat, sort and count them. 

Decompression sickness is one of the main occupational health problems for divers in artisanal fisheries worldwide [5,6,7,8,9]. It can occur when the inert gas (nitrogen) forms intra- or extra-vascular bubbles during ascent due to decreasing atmospheric pressure. These bubbles can obstruct blood flow, damage tissues, and promote the activation of an inflammatory response. Depending on the affected site, mild symptoms, such as joint pain and skin alterations, or more serious neurological and cardiovascular diseases, which are potentially fatal, may occur immediately or even hours after diving. Additionally, when bubbles reach the arterial bloodstream, a risk of arterial gas embolism exists, affecting mainly the brain. Decompression sickness, as well as arterial gas embolism are known as “Decompression Illness” [10,11]. 

Decompression Illness may result in long-term effects on divers’ brains [12] including changes in cognitive function [13]. Brain imaging of patients who suffered decompression illness, especially those with cerebral arterial gas embolism, showed changes in several brain regions, including the frontal and parietal lobes [14,15]. Bubble buoyancy into the bloodstream could explain the location of the lesions [10,12,15,16]. Executive function is one of the cognitive functions circumscribed mainly to the frontal lobe [17,18]. It is understood as complex cognitive processes such as cognitive flexibility, task-shifting, working memory, planning, and inhibition. These processes are interrelated and allow adequate performance in goal-oriented tasks and functioning on a daily basis [19]. Among them, cognitive flexibility is preponderant in adapting behavior to a changing context [20]. Deficits in cognitive flexibility result in perseverative behavior [21], defined as the persistent repetition of a behavior even when it is no longer appropriate [18].

Despite the importance of the executive function in divers’ performance, limited evidence on the effects of decompression illness on executive functioning exists [13,22]. In recent decades, studies of cognitive function—including executive function—focused on assessing long-term effects of diving conditions on the central nervous system of divers with no history of decompression illness [23]. These studies showed poor performance on executive function tests by divers [24] compared to non-divers [25,26]. None of these studies were carried out in artisanal diving. A recent meta-analysis even showed that repetitive diving increased the risk of brain injury in divers with no history of decompression illness [27]. Deep dives and cold environments were factors involved in altering brain perfusion and cognitive performance of divers [28]. In concordance with these findings, we have previously shown a low level of attention in southern Chilean shellfish divers who frequently dived deeper than 30 m [29]. 

Given that decompression illness is frequently occurring among divers in small-scale artisanal fisheries, our objective was to assess frequency and severity of decompression illness as a risk factor for reduced executive functioning among artisanal shellfish divers in Southern Chile.

## 2. Materials and Methods

### 2.1. Study Design and Participants

This cross-sectional study was carried out between September 2017 and October 2018 in two fishing communities located in southern Chile. Seafood and algae harvesting are the main economic activities in these communities. We included male divers between the ages of 18 and 59 years, currently working or having worked in artisanal fishing. Women could not be included as they are not involved in artisanal diving in these communities. We established the upper age limit to avoid severe changes in cognitive function due to aging [30,31,32]. 

In addition, we included a comparison group of never divers. They were fishermen from the same communities, chosen under the same inclusion criteria as the divers. A fisherman was everyone currently working or having worked from the surface of the boats during the harvesting of sea products (e.g., divers’ assistants and crew members of the boats) without a history of diving. 

### 2.2. Field Work and Study Instruments

#### 2.2.1. Recruitment

Prior to field work, we met with social leaders of the artisanal fishing communities (e.g., fishermen’s union) and the villages (e.g., neighborhood councils). We introduced the study to the social leaders and invited the community to participate. After approval by the social leaders, we promoted the study through local radio stations, posters, and flyers at strategic points in each community. For the recruitment of participants, we carried out a door-to-door sampling supported by a local assistant. This recruitment method helped us to recruit past divers and fishermen, and thus, avoid a healthy worker effect. In addition, we thereby were able to include officially registered and non-registered artisanal divers and fishermen. We visited each household up to three times. If an eligible household member agreed to participate, we made a maximum of up to three phone calls to (re-)schedule an appointment (i.e., if the fisherman or diver did not show up for his appointment). 

#### 2.2.2. Data Collection and Data Collection Instruments

Two of the authors (MAG; LM) interviewed the participants and tested executive function. Data collection was done in the participant’s home when conditions were appropriate (i.e., privacy ensured, lighting adequate, and no distractors present). Otherwise, the evaluation was conducted in a room provided by the social community organizations. 

The following study instruments were applied: (1)The Wisconsin Card Sorting Test (WCST) to evaluate executive function, mainly cognitive flexibility and task-shifting [18,33]. In the WCST, the examiner presents a set of 4 stimulus cards on a table. The cards contain figures that differ in shape, color, and quantity. The task of the examinee is to sort response cards with the appropriate stimulus card, according to classification criteria that he must find out. The only instruction he gets is a positive or negative feedback for his classification. The test ends when the examinee manages to solve the task by completing six correct classification categories or when all 128 response cards are finished.(2)An adapted standardized questionnaire to assess socio-demographics, potential confounders [34,35], and work-related exposures using the following sources:-Items developed during our previous work [9,29], to assess sociodemographic data, working and health conditions, diving frequency and depths, and history of decompression illness.-The AUDIT-C questionnaire, corresponding to the first three questions on hazardous alcohol consumption from the World Health Organization’s (WHO) Alcohol Use Disorders Identification Test (AUDIT) [36].-The WHO STEPwise approach to noncommunicable disease risk factor surveillance (STEPS) Instrument [37] to assess smoking behavior.-The PHQ-9 (Patient Health Questionnaire-9), the Chilean validation for the diagnosis of depression in primary health care [38].

In addition, we used an exhaled carbon monoxide monitor (piCO^TM^ Smokerlyzer^®^) to assess possible exposure to carbon monoxide (CO) from the boat engine and the air compressor on the boat as an additional potential occupational source of exposure that might affect brain function [39,40]. Although such potential exposure is outside the pathway between decompression illness and executive function, we evaluated it as a potential risk factor for the outcome. For this, we measured the participants’ exhaled CO (in parts per million- ppm) on-site immediately before and up to 3–5 h after the working day at sea, considering the CO elimination half-time [41]. Despite being a simple procedure, complexity of logistics only permitted to apply this measurement in a subsample of the study population (*N* = 60; 40 divers and 20 fishermen). 

Prior to field work, we conducted a pilot test including 10 fishermen and divers from one of the communities (who did not form part of the study population). The objective was to evaluate the understanding of the questionnaire as well as to verify the feasibility of the procedures, considering the participant’s daily routine. 

#### 2.2.3. Variable Definition

We used three WCST outcome variables to assess executive function: 

(1) Our main outcome was percentage of perseverative responses which corresponds to the proportion of perseverative responses in relation to the number of cards used to complete the test. Perseverative responses mean maintaining one response strategy despite a negative feedback given by the examiner. These responses reflect a poor flexibility in adapting to change [18], i.e., perseverative responses occur when certain perseveration principles are established. For example, when after having correctly completed the “color” sorting category, the individual persists in sorting the following cards by color despite receving negative feedback from the examiner. Greater percentage of perseverative responses suggests less cognitive flexibility.

Perseverative responses mainly result in perseverative errors, but also in some correct responses. In our example, a perseverative error is the persistence in sorting a card according to “color” when the new sorting criterion is e.g., “shape”. Suppose that e.g., the next sorted card has the same shape and color as the stimulus card with which it was paired, and the consecutive card is again sorted by color. In that case, the answer is by chance correct but the response behaviour is still perserverative. 

(2) Our secondary outcome was percentage of perseverative errors which corresponds to the proportion of perseverative errors (see above) in relation to the number of cards used to complete the test. Perseverative errors in the WCST are related to low cognitive flexibility [18,42,43] and shifting ability [18]—“a lower-level form of cognitive flexibility“ [20]—, which are preponderant abilities for performance in the WCST.

(3) As a negative control we used percentage of non-perseverative errors. This measure corresponds to the proportion of errors which are not related to perseverations in relation to the number of cards needed to complete the test [42]. Such errors should not depend on the cognitive flexibility. 

We considered these three WCST variables as percentages instead of absolute values as they have not been validated in the Chilean population. 

Our exposure variable was self-reported history of decompression illness. In absence of a validated measure to assess exposure severity [44], we assigned a score to the history of decompression illness taking frequency, severity, and diagnosis of the condition by a physician into account (Appendix A). A score of (1) was assigned for frequency if the symptoms occurred “once” in a lifetime, (2) if it occurred “several times” and (3) if decompression illness was diagnosed by a physician. The frequency score was multiplied by (1) if mild symptoms were reported, by (2) if it corresponded to neurological decompression sickness, and by (3) in case of cerebral arterial gas embolism. The total score was formed by summing up the single scores. The resulting metric variable was used for the analyses. As a sensitivity analysis, we categorized the score into four groups (0–1 points, 2 points, 3–8 points, and ≥9 points) based on the quartiles of the distribution. The score 0–1 corresponded to divers who did not report any symptoms of decompression illness or who only experienced mild symptoms of decompression sickness once in a lifetime; this group was our internal comparison group. 

We a-priori adjusted the linear and logistic regression models by age assessed in three categories (18–39, 40–49 and 50–59 years), hazardous alcohol consumption (Audit-C score ≥ 5) [45], and depressive symptoms (yes or no). We considered participants as suffering from depressive symptoms (based on the Chilean validation of the PHQ-9) as those reporting anhedonia or low mood “more than half the days” in the last two weeks, meeting or not the other criteria for “major depressive syndrome” or “other depressive syndrome” [38]. 

For descriptive analyses we also presented the educational level in three categories (Incomplete primary, Complete primary & incomplete secondary, and Complete secondary & superior), community (called “A” and “B” herein), and the number of completed categories in the WCST, ranging from zero to six. During the WCST, the participant completed a category by correctly classifying 10 consecutive response cards with the established sorting criterion [46]. We used the number of completed categories as a measure of overall performance on the test [42,43,47]. Finally, we calculated the variation in percentage of exhaled CO as the difference of the absolute values in ppm of exhaled CO after work and the exhaled CO before work (baseline) divided by the baseline value. A positive value thus indicates an increase in exhaled CO over the working day. As smoking largely affects CO levels, we also report the CO results restricted to non-smokers. 

#### 2.2.4. Statistical Analysis

We used the IBM SPSS^®^ Statistics version 25 (IBM, Armonk, NY, USA). Double data entry was performed by two different persons in order to check and correct errors. Data from 13 participants were excluded from analysis due to history of stroke or diagnosis of current psychiatric illness (*n* = 5), or failure/technical problems completing the WCST (*n* = 8). A *p* < 0.05 was considered statistically significant.

We performed Chi-square tests of categorical variables to test for independence between divers and fishermen. Then, we performed non-parametric tests (Kruskal-Wallis test, Independent-Samples Median test, and Spearman’s correlation) to compare the co-variates and the outcome between both groups. We used non-parametric tests due to the relatively small sample size and the skewed distribution of the data. Kruskal–Wallis test was used for those variables with ≥3 categories (age and educational level), while the Independent-Samples Median test was used for dichotomous variables (community, depressive symptoms, and hazardous alcohol consumption). The Spearman’s coefficient was used to test the correlation between the metric variable of history of decompression illness score and each outcome variable. Finally, we performed a linear regression model for each of the three outcome variables, analyzing the relationship between decompression illness score and executive function. The models were adjusted by the a-priori defined potential confounders age, hazardous alcohol consumption, and depressive symptoms. In these models, categorical variables with more than two categories were included as dummy variables. As decompression illness can only occur in divers and not in fishermen, we restricted the linear regression models to the group of divers. 

In a sensitivity analysis, we repeated the linear regression models adding (1) fishermen (defining their history of decompression illness score as (−1)), and (2) adding the educational level as predictor of executive function and (3) removing hazardous alcohol consumption from the model. The latter was done as hazardous alcohol consumption was not associated with executive function in the unadjusted analyses. 

### 2.3. Ethics

The study was approved by the Scientific Ethics Committee of the Valdivia Health Service in Chile, and by the Ethics Committee of the Medical Faculty of the Ludwig-Maxi-milians-Universität München, Germany. Prior to data collection, participants signed an informed consent form. Those participants who could not read or write were assisted by a relative; in this case, the relative’s signature and fingerprint of the participant were recorded on the forms. The study was voluntary and anonymous. To the questionnaires and other records, an ID was randomly assigned which was not linked to the identifying information of the participants. This number was handed out to each participant, in case they later decided to withdraw from the study.

## 3. Results

One hundred sixty-two workers (104 divers and 58 fishermen) participated in our study. The overall response was 62% (64% for divers and 58% for fishermen; Figure 1). Sixty-eight percent of the participants were ≥40 years of age, and 57% had only primary education (Table 1). The overall prevalence of depressive symptoms was 14%, being higher among fishermen (24%) than among divers (9%, pChi2 < 0.01). After excluding smokers (11 fishermen and 13 divers), no statistically significant differences were observed in the variation of exhaled CO between fishermen (median 0%; range 0% to 150%) and divers (67%; −67% to 400%; data including smokers are shown in Table 1). 

Divers reported a median of 150 dives in the last year (min. = 0, max. = 300), of which 83% went to a depth of less than 30 m, and 16% between 30 and 50 m. Seventy five percent of the divers reported mild symptoms of decompression sickness, 37% indicated episodes with neurological symptoms. Twenty divers also reported having had cerebral air gas embolism. Regarding the decompression illness score, 37 of the 47 divers who scored ≥ 3 reported neurological symptoms of decompression sickness and embolism; the remaining 10 were diagnosed with mild decompression sickness.

With respect to the executive functioning, divers and fishermen achieved a median of 2.5 completed categories in the WCST. The median (and ranges) for percentages of perseverative responses, perseverative errors and non-perseverative errors were 22.0 (2–98), 20.0 (2–74), and 21.0 (2–66), respectively. 

There were no statistically significant differences in WCST results between divers and fishermen (Table 2). The percentages of perseverative responses and perseverative errors were higher with increasing age, lower educational level, and if depressive symptoms were present.

Among divers, higher scores of history of decompression illness were associated with higher percentages of perseverative responses and perseverative errors (Table 2). After adjusting for age, depressive symptoms, and hazardous alcohol consumption, the association was confirmed (Table 3 and Appendix A). Likewise, the sensitivity analyses were in accordance with these findings (Appendix A).

## 4. Discussion

We aimed to assess the association between decompression illness and executive function in shellfish divers in southern Chile. While we did not find differences in average executive function between divers and the comparison group of unexposed fishermen, we observed a dose-related decrease in executive function with severity and frequency of decompression illness. This suggests that decompression illness is crucial in long-term effects of diving on the brain. 

To our knowledge, this is the first study assessing WCST perseverations as a measure of executive function in divers with history of decompression illness. In a prospective study, divers with a history of decompression illness had worse results on a memory test than divers without a history of decompression illness. As executive control might be involved in memory test results, this finding is in line with our result [13]. Studies on long-term effects on cognitive functioning of divers without a history of decompression illness further support that diving has negative long-term effects on the central nervous system [23,24,25,26,28]. As in our study, these studies did not find differences in the WCST perseverative responses between divers and less or unexposed comparison groups [23,26]. Therefore, the frequency and severity of decompression illness seems to be a key risk factor for loss of executive function. 

Although further cognitive effects have not been evaluated, brain abnormalities in divers who suffered neurological decompression sickness and cerebral arterial gas embolism were shown [14,15]. Failures in the air supply from the surface, resulting in sudden ascent, triggered cerebral arterial gas embolism in abalone divers who had multiple brain injuries. These injuries were mainly located in the frontal and parietal lobes [15]. Failures in Hookah system are common in small-scale fisheries [5,7] so that our results fit well to these findings. This is of special importance considering that only 37% of the divers who had mild decompression illness and 69% of those who had neurological symptoms received some form of medical care (data not shown). Diagnosis, including assessment of neurological status, early treatment, and follow-up—even in patients with mild symptoms—are crucial for avoiding consequences of decompression illness [10,11,48]. In small-scale fishing communities, access to early diagnosis and treatment is hampered by barriers such as distance to a hyperbaric center [7], funding for treatment, unawareness of proper treatment, and unsafe use of “in-water recompression using air” [8]. In addition, local health care providers frequently lack training in basic handling of divers with decompression illness and hyperbaric centers are frequently lacking or difficult to reach [5,48]. 

While the prevalence of depressive symptoms among divers (9%) was similar to the prevalence of suspected depression among Chilean men (10%), the prevalence among fishermen was higher (24%) [49]. Depression and mental health problems are a common issue among farmers [50,51] and fishermen [52,53,54] in rural communities worldwide. “Contextual community factors” such as employment [51,52], environmental factors (e.g., climate), and poor access to mental health services are involved [51]. Although we cannot establish a causal relationship, this study was carried out just one year after episodes of harmful algae bloom in the study region. This environmental situation affected the work and income of fishermen and their families [55]. In addition, during data collection, the frequent bad weather conditions were unfavorable to recover from the crisis. Therefore, depressive symptoms might be a consequence of these events, since low income has been described as a predictor for depression among fishermen [52]. Re-employment opportunities into the salmon industry to cope with the crisis [56]—mostly for divers due to the industry’s historical demand for qualified divers [55]—might explain the differences in the prevalence of depressive symptoms between fishermen and divers in our study. 

Our study has certain limitations and strengths. Despite the relatively small number of participants in the study, our results could be representative for artisanal diving in Chilean small-scale fisheries, but not for other countries where diving techniques, depths, and diving environment are different. Only few samples of carbon monoxide could be taken; therefore, further analysis of potential chronic exposure and its relationship to the executive function of divers and fishermen is pending. Through door-to-door sampling in the communities we were able to invite a representative sample of fishermen and divers. Thus, we could minimize selection bias and the healthy worker effect. 

To our knowledge, this is the first study in Chilean artisanal divers indicating an association between decompression illness and executive function. Therefore, decompression illness should be addressed by primary prevention measures to prevent cognitive impairment over time. This should best be implemented under a community based approach. Similarly, protective factors from the “cognitive reserve” (e.g., high educational level) that allow the individual to cope with brain deterioration by decreasing the risk of functional impairment (e.g., in dementia) should be considered in prevention and rehabilitation [57,58]. This was confirmed in our sensitivity analyses, indicating a strong relationship between level of education and WCST test performance. 

## 5. Conclusions

In conclusion, our results indicated that decompression illness could be a risk factor for decreased executive functioning of artisanal divers. Prevention of decompression illness, along with early diagnosis and treatment, remains crucial to avoid further damage and should be reinforced in small-scale Chilean fishing communities. Training plans and collaborative networking between health services and social partners are necessary interventions to achieve a better quality of life for workers in artisanal fishing communities. 

## Figures and Tables

**Figure 1 ijerph-18-05923-f001:**
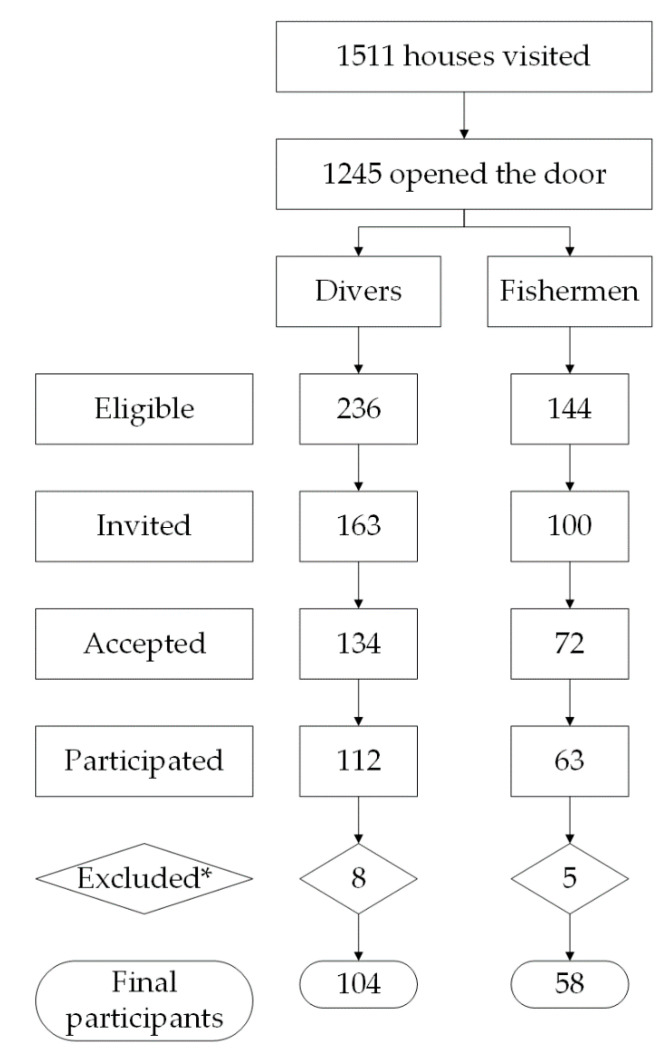
Participants and recruitment process in both communities. * Excluded for (1) having had a stroke or a diagnosis of current psychiatric illness, and (2) failure to complete WCST or records thereof could not be evaluated.

**Table 1 ijerph-18-05923-t001:** Comparison of socio-demographics, potential confounders, variation in percentage of exhaled CO, decompression sickness score, and Wisconsin Card Sorting Test between fishermen and divers.

	Total *N* = 162	Fishermen ^1^ *N* = 58	Divers ^2^ *N* = 104	pChi ^2^ (Divers vs. Fishermen)
	*n* (%)	*n* (%)	*n* (%)	
**Age (years)**				0.05
18–39	51 (31.5)	25 (43.1)	26 (25.0)	
40–49	57 (35.2)	18 (31.0)	39 (37.5)	
50–59	54 (33.3)	15 (25.9)	39 (37.5)	
**Education**				0.61
Incomplete primary	32 (19.8)	10 (17.2)	22 (21.2)	
Complete primary & incomplete secondary	70 (43.2)	28 (48.3)	42 (40.4)	
Complete secondary & superior	60 (37.0)	20 (34.5)	40 (38.5)	
**Community**				0.49
A	133 (82.1)	46 (79.3)	87 (83.7)	
B	29 (17.9)	12 (20.7)	17 (16.3)	
**Depressive symptoms** ^3^	23 (14.2)	14 (24.1)	9 (8.7)	<0.01
**Hazardous alcohol consumption** ^4^	61 (37.7)	26 (44.8)	35 (33.7)	0.16
	N; Median (Min–Max)	N; Median (Min–Max)		*p* (Median test)
**History of decompression illness score** ^5^	n/a	n/a	103; 2.00 (0.00–18.00)	n/a
**% change in exhaled CO** ^6^	60; 0.00 (−80.00–400.00)	20; 0.00 (−40.00–150.00)	40; 33.33 (−80.00–400.00)	0.02
**WCST categories completed** ^7^	162; 2.50 (0–6)	58; 2.00 (0–6)	104; 3.00 (0–6)	0.41

^1^ Former fishermen included in the study *n* = 6. Reasons why they were no longer fishermen: economic (3), change of occupation (1), missing (2). ^2^ Former divers included in the study *n* = 13. Reasons why they were no longer divers: diving accidents (7), other diseases (2), economic (1), missing (2). ^3^ Anhedonia or low mood “more than half the days” in the last two weeks. ^4^ AUDIT-C questionnaire score ≥ 5. ^5^ Score of self-reported history of decompression illness according to severity and frequency of symptoms. Missing *n* = 1. ^6^ Percentage of variation in exhaled carbon monoxide level ((after-before)/before). Smokers are included. Missing *n* total = 102 (64 divers and 38 fishermen). ^7^ Number of WCST completed categories where 0 is the worst and 6 is the maximum.

**Table 2 ijerph-18-05923-t002:** Comparison of Wisconsin Card Sorting Test results by socio-demographics, working conditions, and history of decompression illness (fishermen and divers; *N* = 162).

	% Perseverative Responses		% Perseverative Errors		% Non-Perseverative Errors	
	Median (Min–Max)	*p*	Median (Min–Max)	*p*	Median (Min–Max)	*p*
**Age (years)**		<0.01 _#_		<0.01 _#_		0.94 _#_
18–39	17.0 (2.00–80.00)		16.0 (2.00–63.00)		20.0 (4.00–53.00)	
40–49	22.0 (3.00–98.00)		20.0 (3.00–74.00)		22.0 (2.00–54.00)	
50–59	28.0 (5.00–98.00)		24.0 (5.00–73.00)		22.0 (2.00–66.00)	
**Education**		<0.01 _#_		<0.01 _#_		0.13 _#_
Incomplete primary	32.5 (8.00–98.00)		28.0 (7.00–73.00)		15.0 (2.00–59.00)	
Complete primary & incomplete secondary	22.0 (3.00–97.00)		20.0 (3.00–73.00)		22.5 (2.00–66.00)	
Complete secondary & superior	19.0 (2.00–98.00)		16.5 (2.00–74.00)		20.5 (2.00–53.00)	
**Occupation**		0.29~		0.24~		0.30~
Fishermen	25.0 (2.00–98.00)		22.0 (2.00–74.00)		20.0 (2.00–54.00)	
Diver	21.0 (3.00–98.00)		19.0 (3.00–73.00)		22.5 (2.00–66.00)	
**Community**		0.79~		0.85~		0.25~
B	21.0 (5.00–98.00)		20.0 (5.00–73.00)		19.0 (2.00–54.00)	
A	22.0 (2.00–98.00)		20.0 (2.00–74.00)		22.0 (2.00–66.00)	
**Depressive symptoms** ^1^		0.05~		0.05~		0.20~
No	21.0 (3.00–98.00)		19.0 (3.00–74.00)		22.0 (2.00–66.00)	
Yes	34.0 (2.00–97.00)		29.0 (2.00–73.00)		18.0 (2.00–54.00)	
**Hazardous alcohol consumption** ^2^		0.22~		0.31~		0.60~
No	20.0 (3.00–98.00)		19.0 (3.00–74.00)		22.0 (2.00–66.00)	
Yes	26.0 (2.00–98.00)		22.0 (2.00–73.00)		20.0 (2.00–52.00)	
**History of decompression illness score** ^3^ (Spearman’s coefficient)	0.34	<0.01	0.34	<0.01	−0.17	0.09

_#_ Kruskal-Wallis test, ~ Independent-Samples Median test. ^1^ Anhedonia or low mood “more than half the days” in the last two weeks. ^2^ AUDIT-C questionnaire score ≥5. ^3^ Score of self-reported history of decompression illness according to severity and frequency of symptoms. Missing *n* = 1.

**Table 3 ijerph-18-05923-t003:** Linear regression models to assess the severity and frequency of decompression illness as a risk factor for reduced executive functioning among divers. The models were adjusted for age, depressive symptoms, and hazardous alcohol consumption; *N* = 103 divers with complete data.

	% Perseverative Responses		% Perseverative Errors		% Non-Perseverative Errors	
Adjusted R^2^ (p_Anova_)	0.134 (*p* = 0.002)		0.132 (*p* = 0.002)		0.013 (*p* = 0.284)	
	B	95% CI	B	95% CI	B	95% CI
**Age (years)**						
<40	0		0		0	
40–49	12.95	0.00–25.90	10.06	0.64–19.47	0.01	−6.73–6.75
50–59	14.75	1.34–28.16	10.97	1.22–20.73	3.62	−3.36–10.60
**Depressive symptoms** ^1^ yes vs. no	11.18	−6.17–28.54	7.74	−4.89–20.36	−6.67	−15.71–2.36
**Hazardous alcohol consumption** ^2^ yes vs. no	3.45	−6.81–13.71	2.43	−5.03–9.89	−0.08	−5.42–5.26
**History of decompression illness score** ^3^	1.25	0.20–2.30	0.88	0.12–1.64	−0.46	−1.00–0.09

^1^ Anhedonia or low mood “more than half the days” in the last two weeks. ^2^ AUDIT C questionnaire score ≥ 5. ^3^ Score of self-reported history of decompression illness according to severity and frequency of symptoms.

## Data Availability

The data presented in this study are available on request from the corresponding author. The data are not publicly available due to privacy restrictions.

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
