# Peer review of "Executive Function among Chilean Shellfish Divers: A Cross-Sectional Study Considering Working and Health Conditions in Artisanal Fishing"

_ijerph, 2021, doi:10.3390/ijerph18115923_

Round 1
Reviewer 1 Report
This is a beautiful manuscript evaluating the association between decompression sickness and executive functioning among artisanal divers in southern Chile. based on the frequency and severity of the history of decompression sickness.
A good method of sampling subjects was developed to include in the study by means of door-to-door recruitment by visiting each family up to three times to recruit potential participants.
The limits of the study are to be found in the small number of the sample of participants and in the limited evaluation of a part of the Chilean territory, not including other countries where diving techniques, depths and diving environment are different.
Based on all of this, I absolutely recommend the publication of this manuscript, even if minor changes are required.
The English style should be improved in some paragraphs of the Discussion section by avoiding using too long sentences.
Author Response
Thank you very much for your valuable comments regarding our manuscript! We have followed the suggestions to improve the wording in the discussion, as well as corrected spelling mistakes throughout the text.
Reviewer 2 Report
Thanks for your study on this highly interesting topic. Based on my comments below, I stopped reviewing the paper up and including the results section. I hope the remarks will help you to increase the quality of your work.
Abstract: Remove numerical figures when presenting results as the reader at this stage is not familiar with the scales used.
Line 74: Add a definition/explanation about perseverative behavior.
Line 99: Add references to literature justifying the natural deterioration of cognitive functions as from the age of 60.
Lines 154-155: Define the subset size
Lines 163-174: Explain more clearly the difference between responses and errors. You can include an example.
Lines 188-191:
-The scoring concept seems to ignore that someone could have symptoms more than once but with different severities. How was this accounted for?
-Also, the only scores possible from your approach are: 1, 2, 3, 4, 6, 9. Therefore references to scores of '8' and '>=9' must be excluded. Moreover, as you reasonably assign score 1 as the lowest risk category/comparison group, you must correspondingly declare 9 as the highest risk group. Regarding the rest of the scores, it is not justified why you grouped the '2' separately and '3-6' together. Since you contemplated '2' separately, I would expect also '6' to be assessed separately and the scores '3-4' grouped together.
-Based on the above, you need to reconsider the grouping and rerun your calculations for the particular variable.
Lines 227-242: The overall statistical significance level is not reported. Moreover, as the same data was used for multiple tests, the authors need to apply a correction (e.g., Bonferroni correction).
Table 3:
-It is not clear why you included in your regression alcohol consumption (which was not statistically different in Table 2) and you excluded Education (which was statistically significant).
-The results of linear regressions are not reported fully and correctly, and the model diagnostics are missing.
Author Response
Thank you very much for your comments and feedback, which has helped to improve the quality of the manuscript. Below we answer each point in detail.
Abstract: Remove numerical figures when presenting results as the reader at this stage is not familiar with the scales used.
We have removed the numerical figures as suggested.
Line 74: Add a definition/explanation about perseverative behavior.
We have added a definition to the concept as suggested:
Lines 75-77: Deficits in cognitive flexibility result in perseverative behavior [21], defined as the persistent repetition of a behavior even when it is no longer appropriate [18].
Lines 172 ff: Perseverative responses occur when certain perseveration principles are established. For example, when after having correctly completed the "color" sorting category, the individual persists in sorting the following cards under the same criterion, despite negative feedback from the examiner. This negative feedback should mobilize the examinee to seek the new sorting criterion. Perseverative responses include perseverative errors, but also some correct responses. In our example, a perseverative error is the persistence in sorting a card according to "color" when the current sorting criterion is e.g., "shape". At that point, the principle of perseveration by "color" is established. If we suppose that e.g., the next sorted card has the same shape and color as the stimulus card with which it was paired. In that case, the answer is correct because it has been sorted ac-cording to "shape". However, it also follows the "color" perseveration principle, so it could also be considered as a perseverative response if other necessary conditions are met.
Line 99: Add references to literature justifying the natural deterioration of cognitive functions as from the age of 60.
Thank you, we have included references to justify it:
Lines 102-103: We established the age limit to avoid severe changes in cognitive function due to aging [30-32].
Lines 154-155: Define the subset size
Done as suggested:
Lines 157-159: Despite being a simple procedure, complexity of logistics permitted to apply this measurement only in a subsample of the study population (N= 60; 40 divers and 20 fishermen).
Lines 163-174: Explain more clearly the difference between responses and errors. You can include an example.
Thank you! We have included information and an example to explain it.
Lines 188-191:
-The scoring concept seems to ignore that someone could have symptoms more than once but with different severities. How was this accounted for?
Thank you very much for your comments on the scoring concept. We added table S1 to the paper to better illustrate the concept.
-Also, the only scores possible from your approach are: 1, 2, 3, 4, 6, 9. Therefore references to scores of '8' and '>=9' must be excluded. Moreover, as you reasonably assign score 1 as the lowest risk category/comparison group, you must correspondingly declare 9 as the highest risk group. Regarding the rest of the scores, it is not justified why you grouped the '2' separately and '3-6' together. Since you contemplated '2' separately, I would expect also '6' to be assessed separately and the scores '3-4' grouped together.
Based on the above clarifications, the score ranges from zero to 18. As explained in lines 203/204, groups were built so that their size was comparable.
-Based on the above, you need to reconsider the grouping and rerun your calculations for the particular variable.
We do believe that our groupings were adequate and therefore, did not change the calculations.
Lines 227-242: The overall statistical significance level is not reported. Moreover, as the same data was used for multiple tests, the authors need to apply a correction (e.g., Bonferroni correction).
We have added the overall statistical significance level in our methodology (p<0.05). We however did not consider adjustment for multiple testing as from our point of view, no multiple testing was applied. We tested two hypotheses: 1) there is a difference in cognitive function between fishermen and divers (which was rejected) and 2) cognitive function declines with increasing history of decompression illness (which was confirmed). The remaining variables in the models were potential confounders to be considered in these models.
Table 3:
-It is not clear why you included in your regression alcohol consumption (which was not statistically different in Table 2) and you excluded Education (which was statistically significant).
We a-priori defined the model for the analyses considering potential confounders. Level of education was not among those. We nevertheless conducted a sensitivity analysis including level of education (Table S3). In this analysis, the relationship between higher scores of decompression illness and worse WCST results was maintained.
-The results of linear regressions are not reported fully and correctly, and the model diagnostics are missing.
This information was added to table 3 and table S2/S3.
Round 2
Reviewer 2 Report
Thanks for revising your paper. However, I still have some significant reservations.
-Lines 203-205 and authors' response to the previous relevant comment: Since you have defined a scale, the argument of grouping the scores to achieve equal sample sizes cannot fully stand. This could be the case if you had recorded an interval-type variable (e.g., age) and you wanted to make equal-size groups retrospectively. You need to decide and justify the grouping based on the scale, not the convenience of the sample distribution. The Kruskal-Wallis non-parametric test can deal with the case of unequal sample sizes, and you can always use the Exact options.
-Response to the previous comment on multiple testing and the correction of statistically significant level: Table 2 shows you tested variations of the Wisconsin Card Sorting Test outcomes (your dependent variable) across 7 different groups of variables. This suggests you performed 7 tests with the same sample of the dependent variable. If you performed this test to decide which variables you would consider in the regression test and decide about the order of entering the predictors, then you need to explain this and remove the non-significant variables from your linear regression model. If those tests were included independent from the regression (as it seems) you must apply a correction.
-Although the age and illness score grouping could be logical to adopt to test the differences between fishermen and divers, it is not clear why you do not use their direct values in the tests of variance (e.g., Spearman's correlations in Table 2 instead of Kruskal Wallis) and the linear regression.
-Linear regression:
1. I am afraid I cannot still understand the models presented, while I also did not find all information regarding the outputs and necessary tests per model.
2. The model outputs regarding, the authors must include per model the summary, ANOVA tables, multicollinearity test results (and optionally, casewise diagnostics).
3. The models themselves concerned, I cannot understand the reporting of B and CI values for specific values and not only the variables. For example, Age is one variable/predictor, which could be measured in your model either with values 1, 2 & 3 to correspond with the groups you created or, even better, the exact age per participant (as per my comment above). Why do you report B and CI figures per age group?
4. The predictors entered and their order (which is not also clarified in the paper; was it hierarchical, forced, stepwise?) must be based on some theoretical argument and research results as they affect the model. The fact we collect several data does not mean we must test the model against everything. Your previous tests (Table 2) could be the basis for argumentation here, subject to my respective comments above.
5. In general, this part needs considerable rework.
